# Chloroplast Genome Variation and Phylogenetic Analyses of Seven Dwarf Ornamental Bamboo Species

Binao Zhou [1,†] , Wenjing Yao [1,†] , Chunce Guo [2,†], Lili Bian [1], Yulong Ding [1] and Shuyan Lin [1,*]

1  Co-Innovation Center for Sustainable Forestry in Southern China, Bamboo Research Institute, College of Biology and Environment, Nanjing Forestry University, Nanjing 210037, China
2  Jiangxi Provincial Key Laboratory for Bamboo Germplasm Resources and Utilization, Forestry College, Jiangxi Agricultural University, Nanchang 330045, China
*  Correspondence: lrx@njfu.com.cn
†  These authors contributed equally to this work.

**Abstract:** Dwarf ornamental bamboos are a class of low shrub plants with minor interspecific morphological differences and are difficult to distinguish by traditional classification. In order to identify this type of bamboo species at the molecular level, we sequenced the genomes of the chloroplasts in seven species of dwarf ornamental bamboo: *Pleioblastus argenteostriatus* (Regel) Nakai, *Pleioblastus fortunei* (Van Houtte) Nakai, *Pleioblastus pygmaeus* (Miq.) Nakai, *Pleioblastus pygmaeus* 'Disticha', *Sasaella glabra* (Nakai) Koidz., *Sasaella glabra* 'Albostriata' and *Sasaella kongosanensis* 'Aureostriatus' using high-throughput sequencing. The quadripartite structure of the chloroplast genomes is typical, with sizes ranging from 139,031 bp (*P. argenteostriatus*) to 139,759 bp (*S. kongosanensis* 'Aureostriatus'). The genomes contain 116 genes, including four rRNA genes, 30 tRNA genes and 82 protein-coding genes. Four hotspots, including *ndhI-ndhA*, *trnC-rpoB*, *petB* and *ccsA*, and a total of 46 simple sequence repeats (SSRs) were identified as potential variable markers for species delimitation and population structure analysis. The phylogenetic analyses of chloroplast genomes of seven dwarf ornamental bamboos indicates that these bamboo species can be classified into three categories: *Sasaella* I, *Pleioblastus* II and *Pleioblastus* III. Except *S. kongosanensis* 'Aureostriatus', the other six species were distributed into two branches, indicating that both *S. glabra* and *S. glabra* 'Albostriata' belong to *Pleioblastus* Nakai genus. There are four mutations on the chloroplast genomes of *S. glabra* and *S. glabra* 'Albostriata', suggesting that the mutations may contribute to their obvious different leaf morphologies. Our study reveals the chloroplast structural variations and their phylogenetic relationship and mutation dynamics in seven dwarf ornamental bamboos and also facilitates studies on population genetics, taxonomy and interspecific identification in dwarf bamboo plants.

**Keywords:** dwarf ornamental bamboo; genome variation; simple sequence repeat; interspecific relationships; population genetics



## 1. Introduction

Chloroplast, an important organelle for photosynthesis and energy conversion, participates in the synthesis of secondary metabolites in most higher plants [1,2]. According to studies, the majority of angiosperms' chloroplast genomes have a typical ring-shaped tetrad structure containing a short single-copy region (SSC) and a long single-copy region (LSC) separated by two inverted repeat regions (IRa, IRb) [3]. The length of most chloroplast genomes is 140–160 kb, with less than 180 kb or more than 120 kb [4]. As the chloroplast genome has advantages of structural conservation, lack of recombination from uniparental inheritance, and low degree of sequence variability, it is often used for phylogenetic analysis and species identification compared to nuclear and mitochondrial genomes [5].

Bamboos are remarkably robust forest grasses, of which more than 1400 species in 115 genera have been described in the subfamily Bambusoideae of the family Poaceae [6].

Among the various bamboo species, dwarf ornamental bamboos are low shrub bamboo species with high ornamental and economic value [7]. In addition, dwarf ornamental bamboos have a high reproductive ability and thus can quickly cover the ground after cultivation, which play a great role in ecological balance [8]. The traditional method for the classification of bamboo subfamily is mainly based on the Keng system, in which the inflorescence and rhizome characteristics combined with vegetative characteristics serve as the main characters to divide the genera into the following groups [9–11]. As the flowers and fruits of most bamboo species are difficult to obtain, the species identification of many bamboo species is mostly based on vegetative characteristics [12]. However, the external vegetative organs in bamboos are vulnerable to environmental factors and individual development, and the reliability of bamboo classification is limited to various degrees. Fortunately, this problem has been well solved by genetic markers. As a new kind of genetic marker, molecular markers have many advantages with high reliability, efficiency and precision. Currently, molecular markers, such as simple sequence repeats (SSRs), are used for the studies on origin identification, genetic diversity, interspecific or intergeneric genetic relationship, variety identification and genetic map in bamboo plants [13]. For instance, in the study of Hodkinson et al., the genetic relationship of 20 bamboo species of the genus *Phyllostachys* Siebold et Zucc. was determined by amplified fragment length polymorphism (AFLP) [14].

In this study, we obtained seven dwarf ornamental bamboos with minor interspecific morphological differences among species. The scientific names of bamboo species are from the List of Bamboo Cultivars in China and World Checklist of Bamboos and Rattans [15–17]. Based on the leaf characteristics, the seven bamboo species could be divided into two groups (Figure 1): the leaves with obvious stripes (*Sasaella kongosanensis* 'Aureostriatus', *Sasaella glabra* 'Albostriata', *Pleioblastus argenteostriatus* (Regel) Nakai and *Pleioblastus fortunei* (Van Houtte) Nakai) and the leaves without stripes (*Pleioblastus pygmaeus* (Miq.) Nakai, *Pleioblastus pygmaeus* 'Distichus' and *Sasaella glabra* (Nakai) Koidz.). In particular, the three bamboo species in the second group have relatively similar morphological features which are difficult to distinguish. Thus, we speculated that (1) the classification of *Sasaella glabra* and *Sasaella glabra* 'Albostriata' into *Sasaella* Makino may not be accurate as their reproductive organs have not been observed to date (2) the morphological differences in the leaves between *Sasaella glabra* and *Sasaella glabra* 'Albostriata' possibly results from mutation of chloroplast genome, and (3) *Sasaella kongosanensis* 'Aureostriatus' was incorrectly classified into *Pleioblastus* Nakai before flowering, and was revised into *Sasaella* after its reproductive organs were observed, which can be validated at molecular level. In the study, we profiled the chloroplast genome variation and phylogenetic relationship of the seven dwarf ornamental bamboos to achieve the following objectives: (1) to understand the genome evolution of seven dwarf ornamental bamboos by comparing their chloroplast structure, (2) to determine the highly variable regions for species identification of dwarf ornamental bamboos and (3) to validate the species classification of dwarf ornamental bamboos at the molecular level.

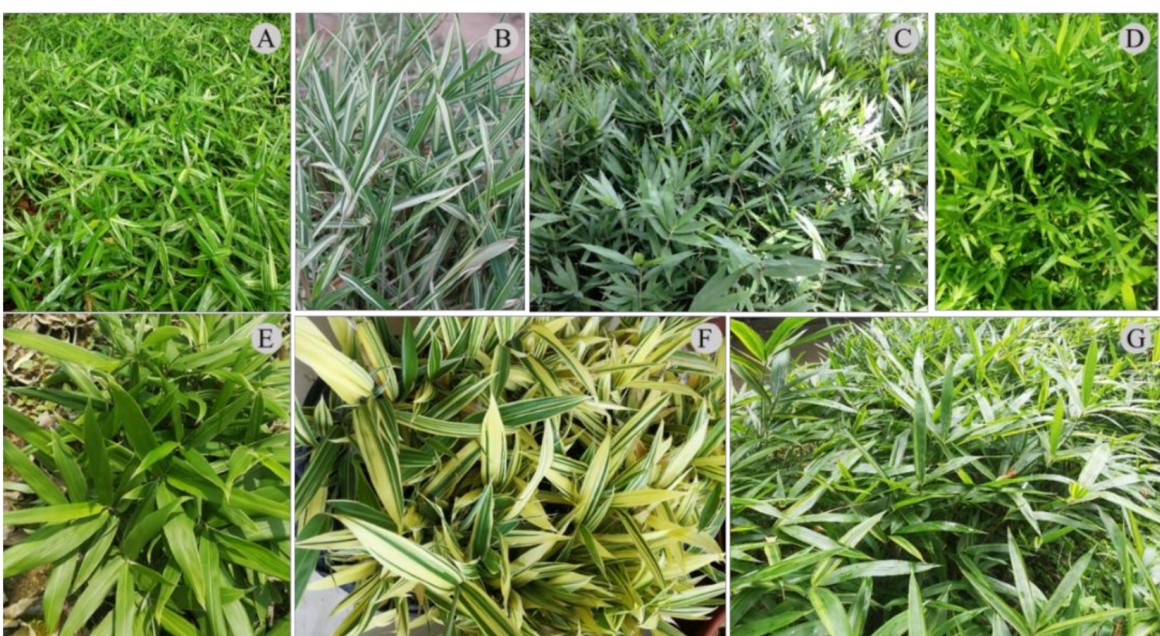

**Figure 1.** Morphological comparison of seven dwarf ornamental bamboos grown in the bamboo garden and Baima experimental base of Nanjing Forestry University, Nanjing, China. (**A**) *Pleioblastus argenteostriatus*. (**B**) *Pleioblastus fortunei*. (**C**) *Pleioblastus pygmaeus*. (**D**) *Pleioblastus pygmaeus* 'Disticha'. (**E**) *Sasaella glabra*. (**F**) *Sasaella glabra* 'Albostriata'. (**G**) *Sasaella kongosanensis* 'Aureostriatus'.

## 2. Materials and Methods

### 2.1. Plant Materials

The seven dwarf ornamental bamboos (*P. argenteostriatus*, NF20210603001; *P. fortunei*, NF20210603002; *P. pygmaeus*, NF20210603003; *P. pygmaeus* 'Disticha', NF20210603004; *S. glabra*, NF20210603005; *S. glabra* 'Albostriata', NF20210603006; *S. kongosanensis* 'Aureostriatus' NF20210603007) were obtained from the bamboo garden of Nanjing Forestry University (118°48′42″ E, 32°04′34″ N) and Baima experimental base of Nanjing Forestry University (119°07′42″ E, 31°37′55″ N), Nanjing. The voucher specimens are deposited in the Dendrological Herbarium, Nanjing Forestry University (NF). Voucher information for samples included in this study is available in Supplementary Table S1. All the seven plant materials used in this study have been previously published or were identified, which can ensure the material source [17,18]. The complete chloroplast genomic DNA of these seven bamboo species was extracted from fresh leaves using modified CTAB method (cetyltrimethylammonium bromide) [19]. The samples were sent to Novogene Co., Ltd. (Beijing, China) for the preparation of DNA library. After quality checking, paired-end sequencing (150 bp) was performed on the Illumina high-throughput sequencing platform Novaseq 6000.

### 2.2. Genome Assembly and Annotation

GetOrganelle v1.7.5.0 was used for genome assembly, with the default parameters. [20]. Bandage was used to visualize the assembled graphs to authenticate the automatically generated chloroplast genomes [21]. Geneious Prime (Version. 2021.2.2) was used to annotate the whole chloroplast genomes with *Phyllostachys edulis* (Carrière) J. Houz. (NC015817), *Pleioblastus amarus* (Keng) Keng f. (NC043892) and *Chimonobambusa sichuanensis* (T.P.Yi) T.H.Wen (NC056904) as references. The annotated genes were checked, and the errors were corrected manually [22]. Additionally, the chloroplast genome sequences of all seven species were submitted to NCBI GenBank with accession numbers OP036432 to OP036438. The circular chloroplast genomic map of these seven bamboo species was drawn and visualized using OG-DRAW online software (https://chlorobox.mpimp-golm.mpg.de/OGDraw.html (accessed on 6 February 2022)) [23].

### 2.3. SSR Analysis

SSR markers from these chloroplast genomes were predicted using MISA online software (https://webblast.ipk-gatersleben.de/misa/ (accessed on 6 February 2022)) [24] with the following parameters: >10 repeat units for mononucleotide, >5 repeat units for dinucleotide, >4 repeat units for trinucleotide, and >3 repeat units for tetranucleotide, pentanucleotide, and hexanucleotide SSRs.

### 2.4. Chloroplast Genome Comparison and Divergent Hotspot Identification

The whole chloroplast genomes of seven bamboos species were compared by mVISTA program (http://genome.lbl.gov/vista/mvista/submit.shtml (accessed on 8 February 2022)) with Shuffle-LAGAN model [25]. The chloroplast genome sequence of *S. kongosanensis* 'Aureostriatus' was used as a reference genome. The nucleotide diversity of these seven chloroplast genomes was calculated based on sliding window analysis using DnaSP (version. 6.12.03) software [26]. The window length was set to 600 bp with step size of 100 bp. The chloroplast genomes of *P. pygmaeus* 'Disticha', *P. fortunei* and *S. glabra* were found to be identical by sequence alignment. Two data sets were created for this analysis: seven dwarf ornamental bamboos as group A and five species excluding *P. pygmaeus* 'Disticha' and *P. fortunei* as group B.

### 2.5. Phylogenetic Analysis

The chloroplast genome sequences of 33 Gramineae species (31 Bambusoideae species and 2 others common Gramineae species) were used for phylogenetic analysis. Among the 31 Bambusoideae species, the information of seven dwarf ornamental bamboo was from our study, twelve from GenBank and twelve from published articles [27] (Table S2). MAFFT was used to align the chloroplast genome sequences, and Gblocks-0.91b was used to remove any ambiguous alignment regions. [28]. Phylogenetic tree was inferred using Bayesian (BI) and maximum likelihood (ML) methods. The optimal model GTR + F + I + G4 + I was calculated by Modelfinder based on BIC standard [29]. ML calculation was performed using IQ-tree with 1000 standard bootstrap [30]. BI was implemented with MrBayes [31]. Markov chain Monte Carlo (MCMC) analysis was performed for 10,000,000 generations. Every 1000 generations, a sample of the tree was taken, and the initial 25% were discarded as burn-in. The average standard deviation of split frequencies was confirmed to be less than 0.01.

## 3. Results

### 3.1. Chloroplast Genome Features

Based on sequencing results, as many as 138,181,858–378,995,026 raw reads were respectively obtained from seven bamboos species (Table 1, Figure 1). The total size of chloroplast genomes ranges from 139,031 bp (*P. argenteostriatus*) to 139,759 bp (*S. kongosanensis* 'Aureostriatus'). The seven chloroplast genomes have a typical tetrad structure, including an LSC region (82,579–83,352 bp), an SSC region (12,815–12,888 bp) and a pair of IR regions (21,796 bp). The mean GC content of the seven chloroplast genomes is 38.9%, and the values are 36.9%–37% in LSC, 33.3% in SSC, and 44.2% in IR (Table 1). There was no obvious difference in the chloroplast genome size among the seven bamboos.

There are 116 genes including four rRNA genes, 30 tRNA genes, and 82 protein-coding genes in each chloroplast genome (Figure 2, Table 2). Among the 116 genes, 11 protein-coding genes (*rps7, rps12, rps15, rps19, rpl2, rpl23, ycf1, ycf2, ycf15, ycf68* and *ndhB*), 8 tRNA genes (*trnH-GUG, trnI-CAU, trnI-GAU, trnL-CAA, trnV-GAC, trnA-UGC, trnR-ACG* and *trnN-GUU*) and 4 rRNA genes are duplicated in the IR regions. Sixteen protein-coding genes (*trnK-UUU, rps16, trnG-UCC, atpF, trnL-UAA, trnV-UAC, petB, petD, rpl16, rpl2, ndhB, trnI-GAU,* trnA-UGC, *ndhA,* trnA-UGC and *trnI-GAU*) contain one intron, while two genes (*ycf3* and *rps12*) contain two introns. The *rps12* is a trans-spliced gene with 5′ end in the LSC region and 3′ end in the IR region. In particular, three protein-coding genes contain

other annotation information; for example, *trnK-UUU, trnI-GAU* and *rrn23* contain *matK, ycf68* and *ycf15*, respectively.

**Table 1.** Features of the complete chloroplast genomes of seven ornamental bamboo species.

| Characteristics | *P. argenteostriatus* | *P. fortunei* | *P. pygmaeus* | *P. pygmaeus* 'Disticha' | *S. glabra* | *S. glabra* 'Albostriata' | *S. kongosanensis* 'Aureostriatus' |
|---|---|---|---|---|---|---|---|
| Unique Reads | 104,781,663 | 267,925,614 | 231,335,814 | 219,217,280 | 106,766,884 | 93,618,968 | 185,056,693 |
| Duplicate Reads | 33,400,195 | 111,069,412 | 85,884,030 | 85,662,068 | 37,603,430 | 35,723,466 | 67,209,637 |
| total reads | 138,181,858 | 378,995,026 | 317,219,844 | 304,879,348 | 144,370,314 | 129,342,434 | 252,266,330 |
| Total genes | 116 | 116 | 116 | 116 | 116 | 116 | 116 |
| Protein-coding genes | 82 | 82 | 82 | 82 | 82 | 82 | 82 |
| rRNA genes | 4 | 4 | 4 | 4 | 4 | 4 | 4 |
| tRNA genes | 30 | 30 | 30 | 30 | 30 | 30 | 30 |
| Total size (bp) | 139,031 | 139,067 | 139,032 | 139,067 | 139,067 | 139,067 | 139,759 |
| LSC length (bp) | 82,579 | 82,587 | 82,580 | 82,587 | 82,587 | 82,587 | 83,352 |
| IR length (bp) | 21,796 | 21,796 | 21,796 | 21,796 | 21,796 | 21,796 | 21,796 |
| SSC length (bp) | 12,860 | 12,888 | 12,860 | 12,888 | 12,888 | 12,888 | 12,815 |
| Total GC (%) | 38.9 | 38.9 | 38.9 | 38.9 | 38.9 | 38.9 | 38.9 |
| LSC of GC (%) | 37 | 37 | 37 | 37 | 37 | 37 | 36.9 |
| IR of GC (%) | 44.2 | 44.2 | 44.2 | 44.2 | 44.2 | 44.2 | 44.2 |
| SSC of GC (%) | 33.3 | 33.3 | 33.3 | 33.3 | 33.3 | 33.3 | 33.3 |
| Accession number | OP036432 | OP036433 | OP036435 | OP036434 | OP036437 | OP036436 | OP036438 |

SSC, short single-copy region; LSC, long single-copy region; IR, inverted repeat regions.

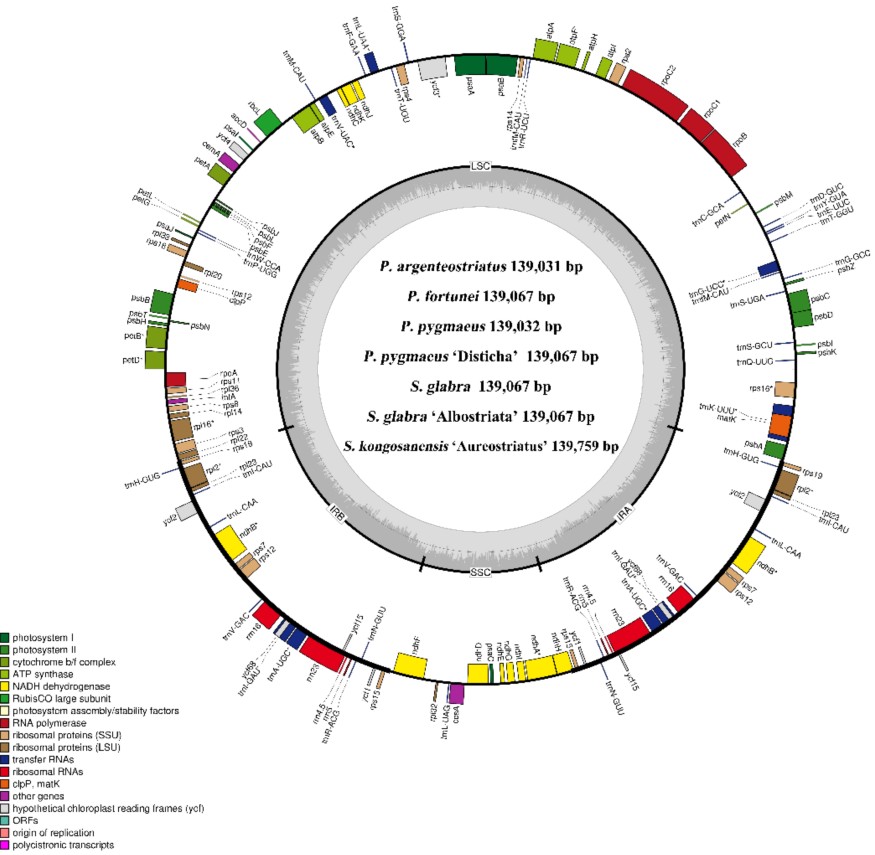

**Figure 2.** The gene map of the chloroplast genomes of seven ornamental bamboo species. Genes on the inside of the large circle are transcribed clockwise, and those on the outside are transcribed counter clockwise. The color-coding of the genes is determined according to their annotation functions. The GC content of the chloroplast genomes is represented by the dashed area.

**Table 2.** List of the annotated genes in the chloroplast genomes of seven ornamental bamboo species.

| Category for Genes | Group of Genes | Name of Genes |
|---|---|---|
| Self-replication | Ribosomal RNA genes | *rrn4.5, rrn5, rrn16, rrn23* |
| | Transfer RNA genes | *trnA-UGC \*, trnC-GCA, trnD-GUC, trnE-UUC, trnF-GAA, trnfM-CAU, trnG-GCC \*, trnG-UCC, trnH-GUG, trnI-CAU, trnI-GAU \*, trnK-UUU\*, trnL-CAA, trnL-UAA \*, trnL-UAG, trnM-CAU, trnN-GUU, trnP-UGG, trnQ-UUG, trnR-ACG, trnR-UCU, trnS-GCU, trnS-GGA, trnS-UGA, trnT-GGU, trnT-UGU, trnV-GAC, trnV-UAC \*, trnW-CCA, trnY-GUA* |
| | Ribosomal protein (small subunit) | *rps2, rps3, rps4, rps7, rps8, rps11, rps12\*, rps14, rps15, rps16\*, rps18, rps19* |
| | Ribosomal protein (large subunit) | *rpl2 \*, rpl14, rpl16 \*, rpl20, rpl22, rpl23, rpl32, rpl33, rpl36* |
| | RNA polymerase | *rpoA, rpoB, rpoC1, rpoC2* |
| | translation initiation factor | *infA* |
| Photosynthesis | Subunits of photosystem I | *psaA, psaB, psaC, psaI, psaJ* |
| | Subunits of photosystem II | *psbA, psbB, psbC, psbD, psbE, psbF, psbH, psbI, psbJ, psbK, psbL, psbM, psbN, psbT, psbZ* |
| | Subunits of cytochrome | *petA, petB \*, petD \*, petG, petL, petN* |
| | Subunits of ATP synthase | *atpA, atpB, atpE, atpF \*, atpH, atpI* |
| | Large subunit of Rubisco | *rbcL* |
| | Subunits of NADH dehydrogenase | *ndhA \*, ndhB \*, ndhC, ndhD, ndhE, ndhF, ndhG, ndhH, ndhI, ndhJ, ndhK* |
| Others | Maturase | *matK* |
| | Envelope membrane protein | *cemA* |
| | Subunit of acetyl-CoA | *accD* |
| | Synthesis gene | *ccsA* |
| | ATP-dependent protease | *clpP* |
| Unknown function | Conserved open reading frames | *ycf1, ycf2, ycf3 \*, ycf4, ycf15, ycf68* |

Intron-containing genes are marked by asterisks (*).

### 3.2. SSRs

*S. kongosanensis* 'Aureostriatus' has 46 SSRs and the other six bamboo species have 45 SSRs (Figure 3). Of the 45 SSRs in the six bamboo species, 36 SSRs loci are located in the LSC region, 4 are located in the IR region, and 5 are located in the SSC region. Additionally, 23, 10 and 12 SSRs are respectively distributed in the spacers, introns and exons. Of the 46 SSRs in *S. kongosanensis* 'Aureostriatus', 37 SSRs loci are located in the LSC region, 4 are located in the IR region, and 5 are located in the SSC region, and 20, 14 and 12 SSRs are distributed in the spacers, introns and exons, respectively. Further analysis indicates that mononucleotide and tetranucleotide repeats account for the most of the SSRs, and mononucleotide repeats are main repeat units with A or T as the predominant ones. Specially, the chloroplast genomes in *S. kongosanensis* 'Aureostriatus' do not have hexanucleotide repeats, while the other six bamboo species do not have pentanucleotide repeats.

### 3.3. Sequence Divergence

To determine the divergence degree of seven dwarf ornamental bamboo species, a comparative analysis of their chloroplast genome sequences was performed based on mVISTA (Figure 4). The chloroplast genome sequences are highly similar and conserved in the seven bamboo species. Particularly, compared to single-copy areas and noncoding regions, IR regions and CDS regions have more conserved sequences. It was found that sequence variations mainly occur in the noncoding regions and large differences exist in the sequences of *S. kongosanensis* 'Aureostriatus' compared to the other six bamboo species. The sequence differences between *S. kongosanensis* 'Aureostriatus' and the other six bamboo species mainly occur in the *trnG-UCC-trnT-GGU, rbcL-accD, ndhC-trnV-UAC, ndhG-ndhI* and *trnS-GCU-psbD* regions. In addition, the genome sequences of *P. argenteostriatus* and *P. pygmaeus* differ from those of *P. fortunei*, *P. pygmaeus* 'Disticha', *S. glabra* and *S. glabra* 'Albostriata' in the *ndhJ-ndhK* and *ndhC-trnV-UAC* regions.

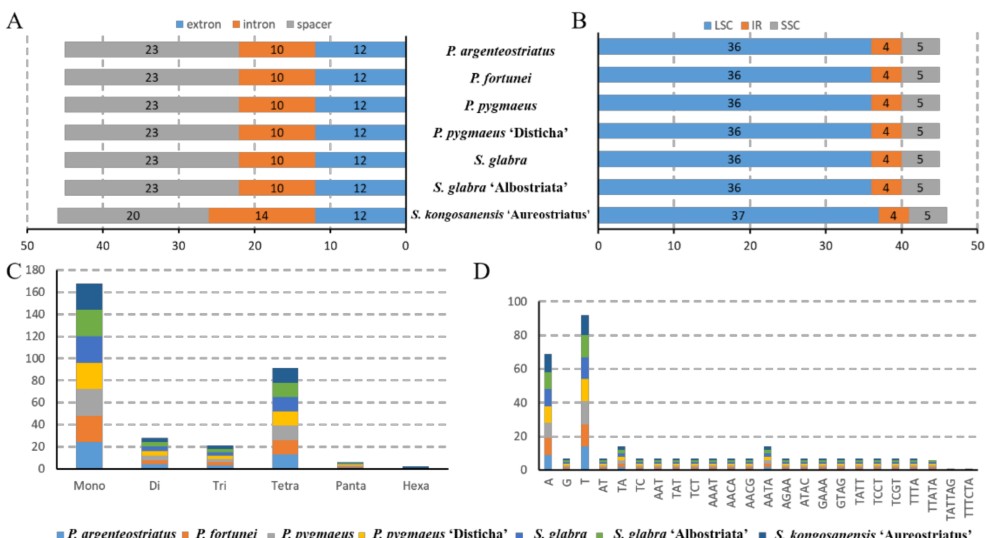

**Figure 3.** The type and distribution of SSRs in the chloroplast genomes of seven ornamental bamboo species. (**A**) Ratio of SSR distribution in different species. (**B**) Frequency of SSR occurrence in the long single-copy region (LSC), short single-copy region (SSC), and inverted repeat regions (IR). (**C**) Number of SSR repeat types. (**D**) Number of identified SSR motifs in different repeat class types.

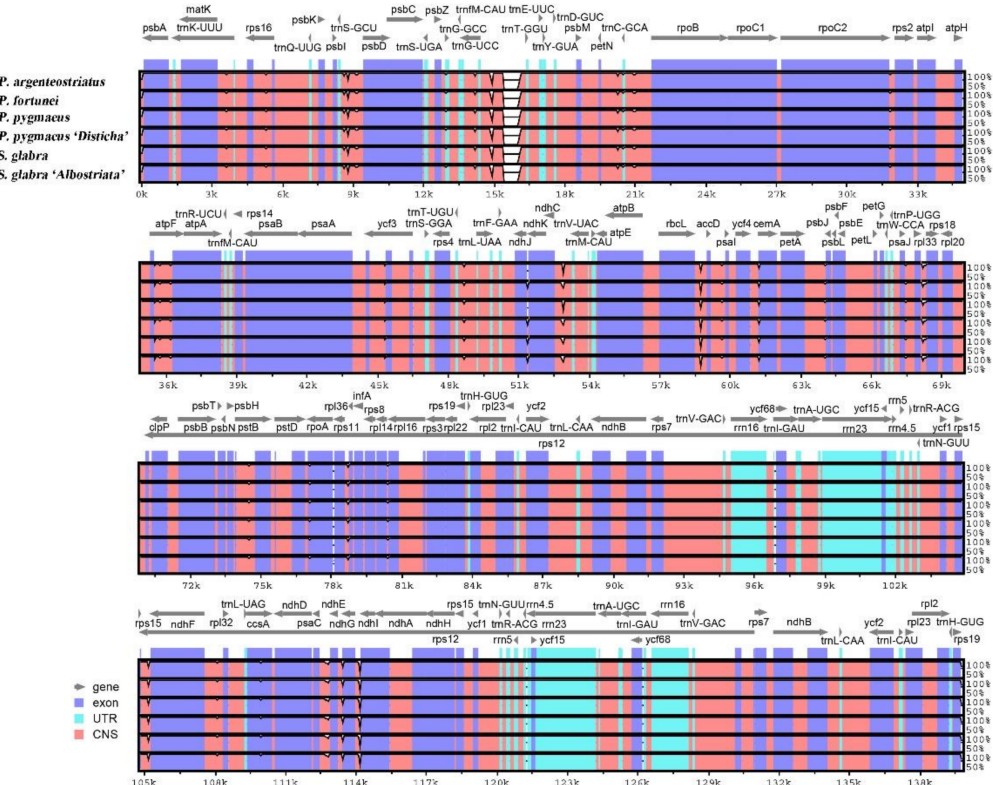

**Figure 4.** Visualization of the chloroplast genome alignment of seven ornamental bamboo species using *S. kongosanensis* 'Aureostriatus' as reference genome by mVISTA. The *x*-axis represents the coordinate of the genes in the chloroplast genome. The aligned regions' sequence similarity is represented by horizontal bars that display the average percent identity within 50 or 100%.

To screen the regions with mutation, the nucleotide diversity of the chloroplast genome was analyzed using DNAsp (Figure 5). Nucleotide diversity values within 600 bp vary from 0 to 0.00524 in group A and 0 to 0.00733 in group B. The region of *ndhI-ndhA* has the highest Pi value (group A = 0.00524, group B = 0.00733), followed by another spacer

region and two gene regions (Pi > 0.005), including *trnC-rpoB*, *petB* and *ccsA*, all of which are located in the single-copy regions such as LSC and SSC.

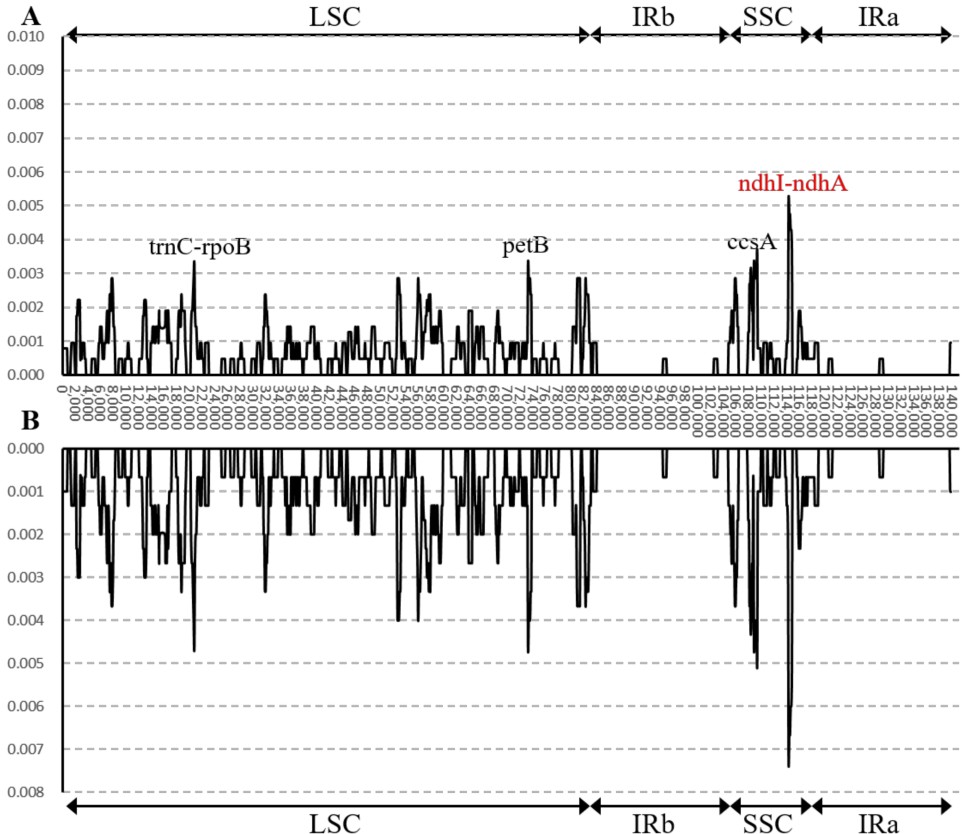

**Figure 5.** Chloroplast genomic analysis using sliding-window method. (**A**) All seven dwarf ornamental bamboo species. (**B**) Five bamboo species, *S. kongosanensis* 'Aureostriatus', *P. argenteostriatus*, *P. pygmaeus*, *S. glabra* and *S. glabra* 'Albostriata'. Window length: 600 bp; step size: 100 bp; *X*-axis: position of the midpoint of a window; *Y*-axis: nucleotide diversity of each window.

Interestingly, both *S. glabra* and *S. glabra* 'Albostriata' have the same size of chloroplast genome, while they display obvious differences in leaf morphology. Further analysis revealed four bp mutations on their chloroplast genome sequences, which are all nucleotide A in *S. glabra* and nucleotide G in *S. glabra* 'Albostriata' (Table 3). The three mutations at sites of 36,274, 41,215 and 50,365 lead to amino acid changes, while the mutation at site of 47,685 has no effect on amino acid sequence.

**Table 3.** The variation in the chloroplast genomes of *S. glabra* and *S. glabra* 'Albostriata'.

| Site | Gene | The base of *S. glabra* | The Base of *S. glabra* 'Albostriata' | The Amino Acids of *S. glabra* | The Amino Acids of *S. glabra* 'Albostriata' |
|------|------|-------------------------|----------------------------------------|--------------------------------|-----------------------------------------------|
| 36,274 | *atpA* | A | G | Thr | Ala |
| 41,215 | *psaA* | A | G | Trp | Arg |
| 47,685 | *trnT-UGU* | A | G | Stop codon | Stop codon |
| 50,365 | *ndhJ* | A | G | Ile | Thr |

### 3.4. Phylogenetic Analysis

In this study, we also constructed ML and BI trees of 31 Bambusoideae species and 2 others Gramineae species based on their whole chloroplast genome sequences, which are essentially consistent with topology (Figure 6). *Zea mays* L. and *Oryza sativa* L. are independently differentiated from the base as outgroups, and the seven dwarf ornamental bamboo species are classified into three clades. *S. kongosanensis* 'Aureostriatus' is divided

into the *Sasaella* clade (I), and the other six bamboo species are divided into the *Pleioblastus* clade. The *Pleioblastus* clade is further divided into two clades (*Pleioblastus* II, and *Pleioblastus* III), the *Pleioblastus* II includes *P. pygmaeus* and *P. argenteostriatus*, and the *Pleioblastus* III includes *S. glabra* 'Albostriata', *S. glabra*, *P. fortunei* and *P. pygmaeus* 'Disticha'.

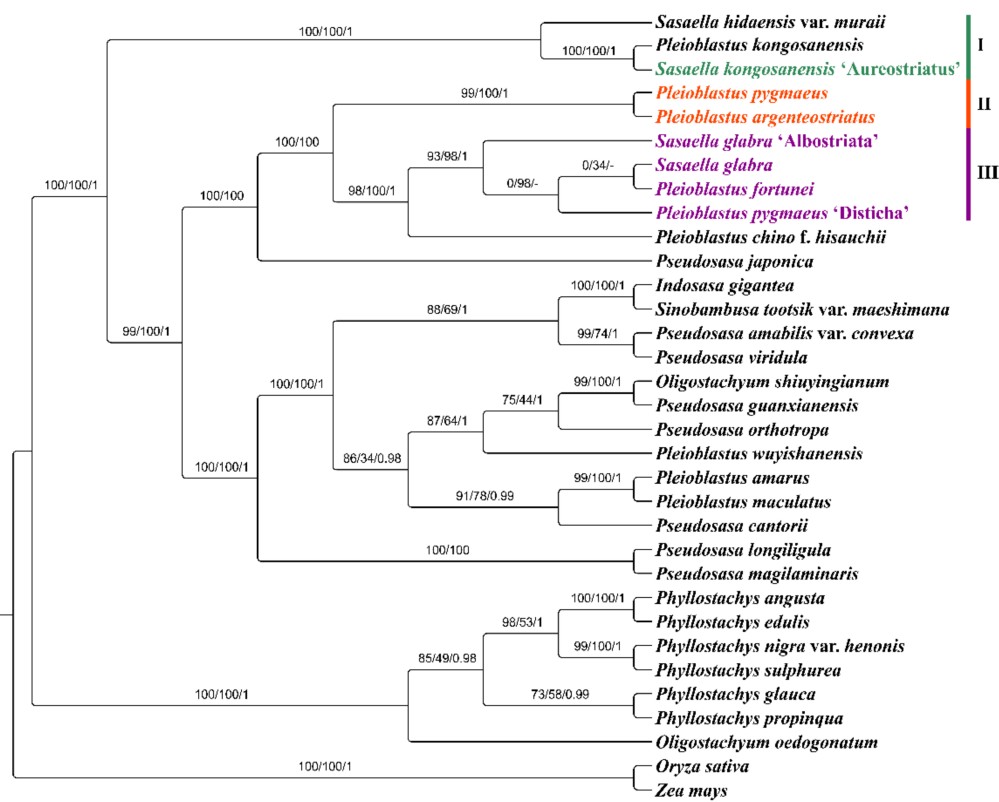

**Figure 6.** Phylogenetic tree of 31 Bambusoideae species and 2 others Gramineae species based on the sequences information of their full-length chloroplast genome using maximum-likelihood (ML) and Bayesian inference (BI) methods. The seven bamboo species in this study are labeled with different colors. The values for the SH-aLRT test, ML bootstrap, and BI posterior probabilities are shown by the numbers above the branches.

## 4. Discussion

In this study, the chloroplast structure and genes are conserved in the seven bamboo species. Every chloroplast genome has the same quadripartite structure and similar genome size of about 139 kb, which is consistent with the size of the bamboo chloroplast genome reported in previous studies [32]. They have a very similar GC content with the highest GC content in IRs. High GC content is beneficial to maintain the genome stability and sequence complexity [33]. In addition, these seven bamboo species have same number of genes and gene classes in the chloroplast genomes, and there is no rearrangement event detected in the genomes. The high similarity in the chloroplast genomes implies that the divergence of the seven dwarf ornamental bamboo species is lower than other species. However, single base substitution phenomena exist in some genes, which has the potential to affect the achievement of gene function and expression. The chloroplast gene *clpP* (ATP-dependent protease) has two introns in *Arabidopsis thaliana* (L.) Heynh. (NC000932) with a length of 1973 bp. Meanwhile, in Poaceae, such as *Oryza sativa* (NC031333), *Zea mays* (NC001666), *Phragmites australis* (Cav.) Trin. ex Steud. (NC022958) and *Triticum urartu* humanjan ex Gandilyan (NC021762), the introns are absent in the *clpP*, and the size is 651 bp in length. The *clpP* of the seven bamboo species in our study, which likewise has no introns, and is 651 bp in length. This may be a common feature in the family Poaceae.

Currently, chloroplast gene markers are easier than nuclear gene markers in studying plant evolution and taxonomy because chloroplast genome is maternally inherited in most plants and its mutation rate is low [34]. Chloroplast genes in bamboo are also maternally inherited, and their mutation rate is low among the seven bamboo species, making it suitable for constructing bamboo phylogenetic trees. Repetitive sequences are important in gene variation and can be used as molecular marker for phylogenetic analysis and population structure [35]. SSR markers are highly repetitive elements with abundant polymorphisms, which are often used in species identification, genetic diversity analysis, gene mapping, and variety purity testing in various plants [36]. However, the phylogenetic relationship of most dwarf ornamental bamboos is limited to date. Therefore, the exploration of additional genetic features in dwarf ornamental bamboo species could improve their phylogenetic resolution. In the present study, a total of 46 SSR markers were detected in the seven bamboo species, which are mostly distributed in the LSC region of the chloroplast and mainly consisted of nucleotide bases A and T. It is consistent with the conclusion that Poly A and poly T are mainly present in SSRs in chloroplast genome sequences [37].

The mVISTA analysis indicates genome variation in the chloroplasts of seven bamboo species. The IR region is less divergent than the LSC and SSC regions, the CDS regions are more conserved, and noncoding regions are more variable, which is consistent with previous chloroplast genome analyses in plants [38]. The large variations were found in the noncoding regions of many genes, such as *trnG-UCC-trnT-GGU*, *rbcL-accD*, *ndhC-trnV-UAC*, *ndhG-ndhI* and *trnS-GCU-psbD*, which could be useful in studying phylogenetic relationship among bamboo species. In addition, the analysis of the nucleotide diversity shows that the pi values of IR regions are much smaller than those of LSC and SSC regions, which is consistent with mVISTA results. Several regions with high pi values were selected, of which the *ndhI-ndhA* region has the highest pi value, which may also provide important insights into species identification and molecular marker development.

As some bamboo species seldom blossom, most species are identified based on their vegetative characteristics [39]. However, several dwarf ornamental bamboo species are similar in morphology, resulting in wrong interspecific taxonomic identification. In this study, the seven dwarf ornamental bamboo species are classified into three classes (*Sasaella* I, *Pleioblastus* II and *Pleioblastus* III) in the phylogenetic tree based on the chloroplast genomes. The original species of *S. kongosanensis* 'Aureostriatus' is *P. kongosanensis*, while there is an incorrect classification of the genus until *S. kongosanensis* 'Aureostriatus' flowered in the bamboo garden of Nanjing Forestry University in 2015. Its flower organs include terminal inflorescence, spikelets with pedicel, rachilla internodes covered with hair, 6 stamens, 1 style, 3 plumose stigmas and caryopsis [40]. The flower morphology is greatly different from the ones of *Pleioblastus* genus, while it is similar with those of *Sasaella* genus. Therefore, *S. kongosanensis* 'Aureostriatus' (OP036438) should belong to *Sasaella* genus, which is also proved in this study. Our results suggest that the chloroplast genome information is more essential and exact for species identification than morphology characters, especially vegetative characters, in bamboo species. Therefore, the phylogenetic relationship of bamboo accessions and variants need to be verified by combined analysis of morphological anatomy and molecular markers in the future.

*S. glabra* and *S. glabra* 'Albostriata' have similar chloroplast genome size and features, while they display different leaf morphology. The chloroplast genomes of the two bamboo species contain four distinct locations, each of which could be allocated to a single gene by sequence comparison. Three mutations in the genes of *aptA*, *psaA*, *ndhJ* change amino acid coding. The *aptA* is the gene encoding α subunits of ATP synthase and the α change in subunit conformation can lead to the breakdown of ATP [41]. The function of *ndhJ* gene is to synthesize NAD(P)H dehydrogenase complexes, thereby reducing plastoquinone (PQ) under light and participating in ring electron transport around PSIto provide additional ATP for photosynthesis under adversity [42]. The mutations of *aptA* and *ndhJ* may affect the physiological and biochemical activities of chloroplasts, thereby enhancing the adaptation

of bamboo plants to stress and ultimately causing variation in leaf structure. The *psaA* is an important gene for photoregulation and can encode P700 apoprotein A, which plays an important role in photosystem I as a primary reaction center in photosynthesis [43]. In a previous study, the expression of *psaA* was higher in the white parts of the leaves than that in the green parts in *Pseudosasa japonica* 'Akebonosuji' [44]. The result indicated that the light transmittance was higher in the white parts of the leaves, strong light can cause oxidative damage of the leaves [45], and the higher expression of *psaA* is beneficial for promoting the synthesis of *psaA* to repair PSI protein complex. For the specific effects induced by mutations in the three genes, we need to perform further physiological and molecular experiments on *S. glabra* and *S. glabra* 'Albostriata' to test our hypotheses. The *P. fortunei* and *S. glabra* are completely identical at the chloroplast genome level. They might have originated from a same maternal line, which contributes to no divergence of chloroplast genomes during lengthy evolution time. The chloroplast sequences of *P. fortunei* and *S. glabra* are the same, but the leaves of *P. fortunei* have stripes. Studies have shown that several factors such as DNA methylation lead to leaf color variation [46,47], so we speculate that the stripes of *P. fortunei* may be related other factors.

## 5. Conclusions

In this study, high-throughput sequencing was used to sequence the chloroplast genomes of seven dwarf ornamental bamboo species, and the seven bamboos have a similar size in chloroplast genome. A total of 116 genes, including 4 rRNA genes, 30 tRNA genes, and 82 protein-coding genes, were found in the chloroplast genomes of all seven species. The genome variation is low in the chloroplasts of seven bamboo species. The IR region is less divergent than the LSC and SSC regions, and the variation among coding regions is less than that of noncoding regions. A total of 46 SSR markers were detected in the seven bamboo species, and these markers are mostly distributed in the LSC region and mainly consisted of nucleotide bases A and T. A phylogenetic tree comprising the seven dwarf bamboo species, 24 other bamboo species and two outgroup plants was constructed based on their whole chloroplast genomes. The seven bamboo species are divided into three branches, and *S. glabra* and *S. glabra* 'Albostriata' are more closely related to the genus *Pleioblastus*, which is inconsistent with their classification based on vegetative characteristics. There are three mutations corresponding to the genes *aptA*, *psaA* and *ndhJ* in the chloroplast genomes of *S. glabra* and *S. glabra* 'Albostriata', which may contribute to different leaf morphology of the two bamboo species. The genetic resources and genomic analysis in this study will facilitate future studies on population genetics, interspecies identification and conservation biology in dwarf ornamental bamboo species.

**Supplementary Materials:** The following supporting information can be downloaded at: https://www.mdpi.com/article/10.3390/f13101671/s1, Table S1: Sample voucher information included in this study; Table S2: NCBI accession number of 33 chloroplast genomes.

**Author Contributions:** S.L. conceived and designed the research. B.Z. conducted data analysis and wrote the manuscript. B.Z., W.Y. and L.B. conducted experiments and data analysis. W.Y., C.G. and Y.D. revised the manuscript. All authors have read and agreed to the published version of the manuscript.

**Funding:** This work was financially supported by the National Key Research & Development Program of China (2021YFD2200503); the National Natural Science Foundation of China (31870595; 32001292); and the Priority Academic Program Development of Jiangsu Higher Education Institutions.

**Data Availability Statement:** The data presented in this study are available in the article and Supplementary Materials.

**Acknowledgments:** The authors thank Muthusamy Ramakrishnan from Nanjing Forestry University for critical reading and editing of the manuscript.

**Conflicts of Interest:** The authors declare no conflict of interest.

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
