# Peer review of "Chloroplast Genome Variation and Phylogenetic Analyses of Seven Dwarf Ornamental Bamboo Species"

_forests, doi:10.3390/f13101671_

Round 1

Reviewer 1 Report

Comments are attached to the pdf.

Author Response

Dear reviewer,

Thank you so much for your careful review and constructive suggestions to improve our manuscript. We have revised the manuscript according to the comments point-by-point. Please see the responses as below.

  1. Materials and Methods
  1. 2.1 - paired-end of how many bp?  Please add the information

Answer: The paired-end is 150 bp, which has been added in the Materials and Methods.

  1. 2.5 - Do you mean you separated all genes, introns, and intergenic spacers and using each region you applied Mafft alignmento and Gblocks trimming?

Answer: We directly used the whole chloroplast genome sequences to build phylogenetic tree, not the partition mode. And the info has been added in the revised manuscript.

  1. 2.5 – the usage of sampling in ML doesn't make sense. We think you mean the ultrafast bootstrap analyses, please correct accordingly.

Answer: Thank you for your suggestion. We used 1000 standard bootstrap in IQtree, which has been revised in the manuscript.

  1. Results
  1. 3.1 - And clpP (?) Doesn't it has 2 introns?

Answer: We annotated the whole chloroplast genomes of seven bamboo species based on those of Phyllostachys edulis (NC015817), Pleioblastus amarus (NC043892) and Chimonobambusa sichuanensis (NC056904). The clpP gene have no introns in all the three referred bamboo species. And we discussed it in the revised manuscript.

  1. 3.1 – Looking at the image it seems that there in something wrong with rps12 annotation (that is why a "brown" part is coloring most of the pt genome. Please correct the annotation file.

Answer: The rps12 is a straddling gene (exons partly in LSC and partly in 2 IRs). We revised it in the annotation file and re-drew Figure 2.

  1. 3.4 – Where in the support values for bayesian inference in these clades?

Answer: The third values are the support values for bayesian inference, which has been presented in the footnote.

  1. Discussion
  1. Discuss the gene clpP in comparison to other plants (exe. other Monocots) Intron quantity for examples

Answer: The clpP has two introns in Arabidopsis thaliana (NC000932) with a full length of 1,973 bp. In Poaceae, such as Oryza sativa (NC031333), Zea mays (NC001666), Phragmites austra-lis (NC022958) and Triticum urartu (NC021762), the clpP has no intron with 651 bp in length. So it may a common feature in the family Poaceae. And we discussed it in the revised manuscript.

  1. Discuss.. Is this and advantage for bamboos?

Answer: Chloroplast genes in bamboo are also maternally inherited, and the mutation rate is low among the seven bamboo species, so it is suitable for constructing phylogenetic trees in the study. And the info has been added in the revised manuscript.

  1. If you increase the sampling, do this statement sustain? If you don't know, please change to "possibly belong to Sasaella"

Answer: Sasaella kogasensis ‘Aureostriatus’ has been classified into the Sasaella genus based on its flower characteristics in our previous study (Lin, S.Y.; Fan, T.T.; Jiang, M.Y.; Zhang, L.; Zheng, X.; Ding, Y.L. The revision of scientific names for three dwarf bamboo species (cultivar) based on the floral morphology. Journal of Nanjing Forestry University(Natural Sciences Edition). 2017, 41, 189-193). Our study verified the classification at molecular level.

Reviewer 2 Report

The manuscript presents interesting genomic resources of seven bamboo accessions and put them in a phylogenetic and comparative framework. While revising it, I had some worries regarding the way some info or data are presented. Then, I made some notes to try to improve or avoid misunderstandings. You will find my comments, suggestions, or issues throughout the PDF file attached here. Although I was able to understand and enjoy the bulk of the manuscript, there are several improvements needed in the writing (especially regarding English).  

Author Response

Dear reviewer,

Thank you so much for your careful review and constructive suggestions to improve our manuscript. We have revised the manuscript according to the comments point-by-point. Please see the responses as below.

Abstract

  1. It is not clear which are these caterogies, and what are they based on. In the next sentence, you mention the mainly "two branches", which I can think as making reference to these categories, but it is still not clear. If you have categories and you're mentioning it, they have to be named, have to be described and recognized. Please, make it clearer.

Answer: Thank you for your suggestions. We have named the three branches as Sasaella I, Pleioblastus II, and Pleioblastus III in the revised manuscript.

  1. I'm not sure if this info is relevant in the abstract. It is also difficult to understand and may be out of context.

Answer: We compared the chloroplast genomes of Sasaella glabra and Sasaella glabra ‘Albostriata’ and found 4 mutations between the two species. We speculate that it may be related to leaf color variation. And we have discussed it in the revised manuscript.

  1. Are you sure about that? I mean: Sasaella glabra and S. glabra 'Albostriata' are nested in a clade with Pleioblastus forunei + P. pygmaeus 'Disticha' (clade III), while Pleioblastus pygmaeus + P. argenteostriatus can be recognized in a distinct clade (clade II). You do not have the type species of Sasaella (S. ramosa) neither of Pleioblastus (P. communis) in your analyses, so you do not know what clade would have priority for genera circumscription. Considering that P. communis (the type species of Pleioblastus) would be embedded in the clade II, you would have reasons to circumscribe that clade as Pleioblastus, and then clade could circumscribe the clade III as Sasaella. You still have uncertainties about the members of clade I. I think you have to be cautious with these taxonomic hypotheses in this case, and I believe your study does not have this goal.

Answer: Thank you so much for your suggestion. In the study, the three bamboo species including Sasaella kogasensis ‘Aureostriatus’, Pleioblastus argenteastriatus, and Pleioblastus pygmaeus have been classified based on their reproductive organs in our previous studies (Lin et al, 2017; Su et al, 2020), with the identification of Sasaella kogasensis ‘Aureostriatus’ classifying into Sasaella genus, Pleioblastus argenteastriatus and Pleioblastus pygmaeus belonging to Pleioblastus genus. The three bamboo species can be regarded as type species to determine the taxonomic status of other bamboo species based on phylogenetic tree. While the reproductive organs of Sasaella glabra and Sasaella glabra ‘Albostriata’ have not been observed to date, their taxonomic status were primarily determined by vegetative traits not flora morphology (Keng, 1996 ). In our study, the two bamboos are grouped into clades III based on phylogenetic tree, which indicates that they may belong to the Pleioblastus genus. Their taxonomic status needs to be determined by flora morphology in the future.  

Lin, S.Y.; Fan, T.T.; Jiang, M.Y.; Zhang, L.; Zheng, X.; Ding, Y.L. The revision of scientific names for three dwarf bamboo species (cultivar) based on the floral morphology. Journal of Nanjing Forestry University(Natural Sciences Edition). 2017, 41, 189-193. https://doi.org/10.3969/j.issn.1000-2006.2017.01.029

Su, J.L.; Shi, W.S.; Yang, Y.Y.; Wang, X.; Ding, Y. L.; Lin, S.Y. Comparison of Leaf Color and Pigment Content and Observation of Leaf Structure at Different Growth Stages from Six Bamboo Species. Scientia Silvae Sinicae, 2020, 56(07):194-203.

Keng, P.C.; Wang, Z.P. Flora of China. Science Press, Beijing, China, 1996, Volume 3

  1. What species are you referring to?

Answer: The two species are S. glabra and S. glabra ‘Albostriata’. We have modified it in the revised manuscript.

  1. Introduction
  1. I'm not so familiar with Bambusoideae taxonomy, so I've made a quick literature revision. I've found mention to "Geng system", but it seems to be more recognized as "Keng system". Please, look at Clark et al. (2015, "Bamboo taxonomy and habitat" - check full reference below). Also, I'm not sure what are you referring to "nutrients characteristics". Is this related to the soil/nutrient requirements of the species, such as some species having habitat preferences related to the type of soil/nutrients? Or is this related to some nutrients that some species accumulate in their bodies/cells? Please, make it clearer to what you are referring to. You can also include a more comprehensive literature citation for your sentences regarding the taxonomy system in China - see references in Clark et a. 2015.

Clark, L.G., Londoño, X., Ruiz-Sanchez, E. (2015). Bamboo Taxonomy and Habitat. In: Liese, W., Köhl, M. (eds) Bamboo. Tropical Forestry, vol 10. Springer, Cham. https://doi.org/10.1007/978-3-319-14133-6_1

Answer: Thank you for your suggestion. We have corrected "Geng system" to "Keng system" and added the reference (Clark et al., 2015) in the revised manuscript. And “Nutrients characteristics” has been changed to “vegetative traits” in the revised manuscript. The vegetative organs of bamboo vary with climate, soil nutrients, and other environment factors. Therefore, it is not accurate to determine the taxonomic status of bamboo species based on vegetative traits. For example, Sasaella glabra and Sasaella glabra ‘Albostriata’ were classified into Sasaella genus based on vegetative traits. However, they were grouped into Pleioblastus III based on phylogenetic tree.

  1. Methods
  1. 2.1 - It is extremely important to have more info regarding the accessions you used. They must be cited with an herbarium voucher, preferentially in an indexed herbarium - if possible - for each accession. It's also not clear how you defined/identified each accession in their taxonomic unit (species). Have you used some taxonomic treatment, dichotomous key, literature, or just the info from the gardens you accessed? In any case, you have to mention it. Further, please, make it clearer which method you used for DNA extraction, which plant part you used, and which method for sequencing (genome skimming, hybrid capture, etc.), Illumina platform, and sequence length

Answer: In the study, the three bamboo species including Sasaella kogasensis ‘Aureostriatus’, Pleioblastus argenteastriatus, and Pleioblastus pygmaeus have been classified based on their reproductive organs in our previous studies (Su et al, 2020; Lin et al, 2017). The three bamboo species can be regarded as type species to determine the taxonomic status of other bamboo species based on phylogenetic tree.

Leaf DNA was extracted by CTAB method and DNA sequencing was performed by Illumina platform. The detailed information has been added in the revised manuscript.

  1. 4 - I think at least one of the three referred 'species' should be kept in this group B, then you will have the total 5 species in the analysis.

Answer: Thank you for your suggestion. We kept Sasaella glabra in the group B, and re-drew Figure 5B based on the total 5 species in the revised manuscript.

  1. Results
  1. 3.2 - This description is confusing and needs to be rephrased. I think you do not need to classify this variation of SSR as "two types". The only distinct one is S. kogasensis 'Aureostriatus". Further, you use "Type A" and "B" to describe them as different, but you have in the Fig. 3 the graphs A and B, and I promptly checked the figure as they were referring to each of the putative "types".

Answer: Sorry for the confusing description. As you suggested, we conducted SSR variation analysis as whole in the revised manuscript.  

  1. 3.4 – Consistent with what? Bootsrapp/posterior probability values? Topology? You should mention about your support values in this paragraph.

Answer: The phylogenetic trees are topologically consistent based on ML and BI methods. The information has been added in the revised manuscript.

  1. 3.4 – Which one is the Sasaella clade? Check my comment in the abstract.

Answer: We have named the three branches as Sasaella I, Pleioblastus II, and Pleioblastus III in the revised manuscript. The Sasaella I includes Sasaella hidaensis var. muraii, Pleioblastus kongosanensis and Sasaella kogasensis ‘Aureostriatus’.

  1. 3.4 – Which one is the Pleioblastus clade? You also have "Pleioblastus" taxa (P. maculatus, P. amarus, P. wuyishamensis) in other clades, below represented... Check my comment in the abstract.

Answer: Pleioblastus clade includes Pleioblastus II and Pleioblastus III. In this study, two Pleioblastus genus species including Pleioblastus argenteastriatus and Pleioblastus pygmaeus have been previously identified by their flora characteristics (Lin et al, 2017), which can ensure clade II and III belong to Pleioblastus genus. However, the taxonomic status of a few Pleioblastus taxa distributed in other clades is not very accurate at present. For example, P. wuyishamensis in other clade was not classified based on vegetative traits but not flora characteristics.

Discussion

  1. Please, be more specific: are conserved in which features? Structure, gene content, size... ?

Answer: In this study, the structure and gene content of the sequenced chloroplast genomes are conserved in the seven bamboos, which is similar to the majority of other bamboo plants.

  1. That's interesting. Usually, GC content is higher in coding (CDS) regions. You can explore a little more about this in your discussion.

Answer: Studies have shown that chloroplast genomes of most angiosperms have a typical ring-shaped tetrad structure containing a short single copy region (SSC) and a long single copy region (LSC) separated by two inverted repeat regions (IRa, IRb). The rRNA gene is located in IRs, its high GC content is conducive to genome stability and the complexity of maintaining sequences. The info has been added in the revised manuscript.

  1. Please, revise this sentence, as it is not making sense with the previous and next phrases.

Answer: Thank you for your suggestion. We agree with you and deleted the sentence in the revised manuscript.

  1. I would suggest to change the writing perspective. Insted of emphasizing what is lower, talk about what is higher: CDS regions are more conserved, noncoding regions are more variable, IR region is more "divergent".

Answer: Thank you for your suggestion. We have revised the manuscript as your comment.

  1. I've read all the paper so far, and have not been meaningfully presented to these "three classes". So, I'm not sure about that.

Answer: To make it clearer, we have named the three branches as Sasaella I; Pleioblastus II; Pleioblastus III in the revised manuscript.

  1. I could not understand this phrase.

- What do you mean with "original species"? The basionym? Or are you refering to your accession, used in your study and represented in the phylogeny (Fig. 6)?

- Additionally, the second part is not clear; what is the relation with a flowering bamboo species in the garden with the taxonomic observations you are doing?.

Answer:

- Species (Original species) is the plant species with original characteristics. Varieties undergo variation and have specific stable characteristics, compared to original species. For example, Sasaella glabra is original species, and Sasaella glabra ‘Albostriata’ is a variety of Sasaella glabra. 

- The vegetative organs of bamboo vary with climate, soil nutrients, and other environment factors. Therefore, it is accurate to determine the taxonomic state of bamboo plants by flora characteristics. The flowering bamboo species can be regarded as type species to determine the taxonomic status of the bamboo species lack of flowering information based on phylogenetic tree.

  1. Now I could understand your point. Please, revise your previous sentence to make it clearer. Maybe if you make mention to your accessions instead of using the taxonomic names would help.

Answer: Thank you for your suggestion. We have added the accession information of S. kogasensis ‘Aureostriatus’ in the revised manuscript.

  1. Why? Based on what? Please, bring more info about your speculation.

Answer: The chloroplast sequences of P. fortune and S. glabra are exactly same, but their leaves display different morphology. Studies have shown that several factors such as DNA Methylation lead to leaf color variation (Shi, et al, 2020; Li, et al, 2011), so we speculate that the stripes of P. fortune may be related other factors. We have added the info in the revised manuscript.

Shi J.; Xiong Y. J.; Chen H.L.; Fang X.Z.; Ling J.; Liu T.L.; Zhang M.M.; Yu J.S.; Xue T.; Xue J.P. Variation Analysis of Genomic DNA Methylation in Photinia Frasery Leaves during Color Transition. Journal of Huaibei Normal University(Natural Sciences), 2020, 41, 47-52.

Li T.C.; Fan H.H.; LI Z.P.; Wei J.; Cai Y.P.; Lin Y. Effect of different light quality on DNA methylation variation for brown cotton(Gossypium hirstum), African Journal of Biotechnology, 2011, 33, 6220-6226.